# Influenza vaccination uptake and factors influencing vaccination decision among patients with chronic kidney or liver disease

**Michael Eder**[1⊙], **Haris Omic**[1⊙], **Jana Gorges**[1], **Florian Badt**[1], **Zeljko Kikic**[1,2], **Marcus D. Saemann**[3], **Allison Tong**[4,5], **David Bauer**[6,7], **Georg Semmler**[6,7], **Thomas Reiberger**[6,7], **Heimo Lagler**[8‡], **Bernhard Scheiner**[6,7‡]*

**1** Division of Nephrology and Dialysis, Department of Medicine III, Medical University of Vienna, Vienna, Austria, **2** Department of Urology, Medical University of Vienna, Vienna, Austria, **3** Department of Nephrology, Clinic Ottakring, Vienna, Austria, **4** Sydney School of Public Health, University of Sydney, Sydney, New South Wales, Australia, **5** Centre for Kidney Research, The Children's Hospital at Westmead, Westmead, New South Wales, Australia, **6** Division of Gastroenterology and Hepatology, Department of Medicine III, Medical University of Vienna, Vienna, Austria, **7** Vienna Hepatic Hemodynamic Lab, Medical University of Vienna, Vienna, Austria, **8** Division of Infectious Diseases and Tropical Medicine, Department of Medicine I, Medical University of Vienna, Vienna, Austria

⊙ These authors contributed equally to this work.
‡ These authors also contributed equally to this work.
* bernhard.scheiner@meduniwien.ac.at

**Data Availability Statement:** After counselling the responsible data clearing committee (Daten-Clearingstelle der Medizinischen Universität Wien, Spitalgasse 23, 1090 Wien,

## Abstract

### Introduction

Seasonal influenza is a major global health problem causing substantial morbidity and health care costs. Yet, in many countries, the rates of influenza vaccination remain low. Chronic kidney or liver diseases (CKLD) predispose patients to severe influenza infections, but data on vaccination acceptance and status is limited in this risk population. We investigated the influenza vaccination awareness considering sociodemographic factors in CKLD patients.

### Patients and methods

This cross-sectional, questionnaire-based study recruited CKLD patients managed at three Viennese tertiary care centers between July and October 2020. CKLD was defined as chronic kidney- (all stages) or compensated/decompensated liver disease, including kidney/liver transplant recipients. Questionnaires assessed sociodemographic and transplant-associated parameters, patients vaccination status and the individuals self-perceived risks of infection and associated complications.

### Results

In total 516 patients (38.1% female, mean age 56.4 years) were included. 43.9% of patients declared their willingness to be vaccinated in the winter season 2020/2021, compared to 25.4% in 2019/2020 and 27.3% in 2016–2018. Vaccination uptake was associated with the self-perceived risks of infection (OR: 2.8 (95%CI: 1.8–4.5), p<0.001) and associated

datenclearing@meduniwien.ac.at, www.
meduniwien.ac.at/daten-clearingstelle), we are now
able to share the anonymized data set. However,
according to the data clearing committee and in
line with the PLOS Data Sharing policy, the data
may only be shared upon request as these data
contain potentially identifying information. Data
access requests may be directed to the office of the
Division of Gastroenterology and Hepatology,
Department of Internal Medicine III, Medical
University of Vienna either per e-mail (gastro-
sekretariat@meduniwien.ac.at) or by post (Division
of Gastroenterology and Hepatology, Department
of Internal Medicine III, Sekretariat, Leitstelle 7i,
Währinger Gürtel 18-20, 1090 Wien).

**Funding:** The authors received no specific funding
for this work.

**Competing interests:** The authors have no
competing interests.

**Abbreviations:** ACLD, advanced chronic liver
disease; CKD, chronic kidney disease; CKLD,
chronic kidney or liver disease; COVID-19,
Coronavirus disease 2019; HD, haemodialysis;
IQR, interquartile range; LTX, liver transplantation;
RTX, renal transplantation; SD, standard deviation;
SOT, solid organ transplantation.

complications (OR: 3.8 (95%CI: 2.3–6.3), p<0.001) as well as with previously received influ-
enza vaccination (2019/2020: OR 17.1 (95%CI: 9.5–30.7), p<0.001; season 2016–2018:
OR 8.9 (95%CI: 5.5–14.5), p<0.001). Most frequent reasons for not planning vaccination
were fear of a) graft injury (33.3%), b) complications after vaccination (32.4%) and c) vac-
cine inefficiency (15.0%).

## Conclusion

While influenza vaccination willingness in patients with CKLD is increasing in the 2020/2021
season, vaccination rates may still remain <50%. Novel co-operations with primary health
care, active vaccination surveillance and financial reimbursement may substantially improve
vaccination rates in high-risk CKLD patients.

## Introduction

Seasonal influenza is a global health problem causing substantial morbidity and mortality as
well as high costs for health care systems worldwide [1–3]. The majority of infections occurs in
children, but severe courses are mainly observed in very young- as well as elderly- or immuno-
compromised individuals [4]. According to the Austrian Federal Health Ministry, influenza
vaccination is recommended for all adults and particularly for persons aged >60 years,
patients with chronic illnesses or other risk factors as well as health care workers [5]. Addition-
ally, influenza vaccination was recently included into the cost-free children vaccination pro-
gram and is generally recommended in children aged >6 months. Despite these extensive
national influenza vaccination recommendations, a comparably low vaccination rate of about
6% (2015/16) in the general Austrian population compared to about 73% in the UK was
recently reported [6]. Suggested reasons include the lack of financial reimbursement and social
marketing, lack of continuity and separated specialty care within the health care system, nega-
tive attitudes of health care workers along with generally low vaccination rates in adults [7].
While higher influenza vaccination rates have been reported among some specific populations,
including patients with rheumatic or malignant diseases in Austria, fear of side effects or wors-
ening of the primary disease remains one of the most intriguing obstacles to increase the will-
ingness for vaccination [8, 9].

Patients with chronic kidney or liver disease (CKLD) comprise a particularly vulnerable
patient population as multiple studies have reported increased risks for severe clinical courses
[10–15]. Therefore, scientific guidelines recommend annual influenza vaccination for patients
with chronic kidney- [16] or chronic liver disease [17, 18] as well as for solid organ- and
hematopoietic stem cell transplant recipients [19, 20]. Similarly, the Austrian Federal Health
Ministry recommends vaccination against s. pneumoniae for all children as well as all adults
aged >60 years. Vaccination is further strongly recommended in patients with a high risk for a
severe course of disease such as patients with chronic kidney disease, liver cirrhosis as well as
patients receiving solid organ transplantation [5]. Vaccination against h. influenzae B (HiB) is
generally recommended for children as well as for patients with immune deficiencies such as
patients with deficiencies in T- and B-cell function [5]. Additionally, HiB vaccination is rec-
ommended in selected patients receiving solid organ transplantation [21]. Data on vaccination
rates, the awareness of the critical importance of vaccination in general as well as on specific
aspects in these special populations are required to improve patient care but have not been

reported in Austria yet. The aim of our study was to evaluate (i) the influenza vaccination uptake in the upcoming winter season 2020/2021 and (ii) to analyze factors influencing decision-making for having an influenza vaccine by assessing sociodemographic- and transplant associated parameters as well as subjective individual reasons among patients with CKLD.

## Materials and methods

### Study design

This cross-sectional survey was conducted at three departments at two hospitals in Vienna, Austria. (Department for Nephrology and Department for Gastroenterology and Hepatology, Medical University of Vienna and Department for Nephrology, Clinic Ottakring, Vienna, Austria). Our primary objective was to evaluate the willingness to receive influenza vaccination in kidney- or liver transplant recipients and patients with CKLD in the upcoming winter season 2020/2021. The secondary objectives were to evaluate vaccination awareness by asking about patients self-assessed influenza infection and complication risks as well as subjective experiences influencing vaccination behaviour. This study was approved by the institutional ethics committees of the Medical University of Vienna (No. 1465/2020) as well as the city of Vienna (20-215-VK). The need for written informed consent was waived by the Ethics committees, as data was acquired in a completely anonymized way.

### Participants eligibility and recruitment

Adult patients (aged ≥18 years) after kidney- or liver transplantation, patients with diagnosed chronic kidney diseases of all stages (including patients undergoing haemodialysis) or advanced chronic liver disease (ACLD) were eligible to participate. Patients undergoing peritoneal dialysis were not included. ACLD was defined by advanced fibrosis or cirrhosis (F3/F4) according to liver histology, a hepatic-venous pressure gradient ≥6mmHg or a transient elastography of ≥10kPa [22]. We used a consecutive purposive strategy and approached patients during visits at the outpatient department, ward or dialysis unit between July and October 2020.

### Questionnaires

The survey was developed based on existing literature on awareness and uptake of vaccination [8, 9] and a discussion among a multidisciplinary team. The printed questionnaire was completed anonymously. Patients entering the nephrology/hepatology outpatient clinic or dialysis units were screened for study inclusion/exclusion criteria by attending physicians, nurses or non-medical hospital staff. If patients fulfilled inclusion criteria and did not fulfil exclusion criteria, questionnaires not asking for any personal data potentially allowing patient identification (i.e. name or date of birth) were distributed. Patients were asked to answer the written questionnaire alone. Afterwards, patients were instructed to return the anonymously completed questionnaire before leaving the hospital. The questionnaire consisted of nineteen questions evaluating demographic (age, gender, country of origin, education level and marital status) and transplant-associated (donor type, age at transplantation, number of previous transplantations) parameters as well as the current and previous influenza vaccination status (including vaccination status in the season 2019/2020 and 2016–2018). The main endpoint was willingness to be vaccinated against influence in the winter season 2020/2021 by asking: "Are you planning to get vaccinated against influenza for the next winter season 2020/2021?" (as demonstrated in the questionnaire added as S1 Table). Patients were asked about factors considered for their decision to have the influenza vaccination performed, which included the

following: medical recommendation against/for influenza vaccination, source of recommendation for vaccination (general practitioner, specialist for nephrology or hepatology, nurse, transplant outpatient clinic, dialysis unit), individual motives (previous influenza infection, self-assessed knowledge, previously experienced side effects, contraindications, interactions, possible harms to transplanted organ), logistical reasons (availability of vaccination, affordability, time requirement for vaccination), source of possible recommendation against vaccination and the patients self-assessed risk of influenza infection and severe disease courses. Furthermore, patients were asked if they received vaccination against pathogens associated with a higher risk of a severe disease course in post-transplant- or immunosuppressed populations such as streptococcus pneumoniae or haemophilus influenzae.

## Statistical analysis

Continuous variables were expressed either as mean ± standard deviation (SD) or median and interquartile range (IQR), as appropriate. Categorical variables are reported as absolute and relative frequencies. For comparison of categorial parameters such as sex, highest school degree or donor type, chi-squared test was used. Normally distributed continuous parameters such as age were compared using unpaired T-test, non-normally distributed parameters were compared with the Mann-Whitney U Test. Distribution of data was evaluated by data visualization using histograms. We further calculated odds ratios via binomial logistic regression to analyze associations between the primary outcome: "willingness to receive influence vaccination in the 2020/2021 season" and reported questionnaire items. Significant associations were then included into a multivariable regression model. A two-sided p-value <0.05 was considered statistically significant. SPSS for Windows, Version 26 (SPSS Inc., Chicago, IL, USA) was used for statistical analyses.

# Results

## Study cohort

In total 516 [38.1% female, mean age 56.4±14.9 years (mean±SD)] patients agreed to participate and were included into the analysis (baseline characteristics are provided in Table 1). Most patients (N = 293/56.8%) were kidney transplant recipients (N = 276/94.8% with their first allograft), followed by 95 (18.4%) patients receiving chronic hemodialysis [23] and 73 CKD patients (14.1%) not receiving kidney replacement therapy. Ten (1.9%) were liver transplant recipients and 45 (8.7%) patients had ACLD. Most participants had completed vocational training / apprenticeship (N = 224/44.2%) and high-school education (N = 98/19.3%). Nine patients (1.7%) did not report the level of education. In total, 107 patients (21.5%) were not born in Austria (most frequent native countries: Serbia N = 16, Turkey N = 13 and Poland N = 10; country of origin not indicated N = 18). The participant characteristics are provided in Table 1.

## Vaccination status and uptake

43.9% of patients declared their willingness to be vaccinated in the winter season 2020/2021, compared to an actual vaccination rate of 25.4% in 2019/2020 and 27.3% in 2016–2018. Patients who received influenza vaccination in the previous seasons were significantly more likely to declare their willingness to take influenza vaccination in the upcoming winter season 2020/2021 [odds ratio (OR) for patients who received influenza vaccination during the season 2019/2020: 17.1 (95%CI: 9.5–30.7), p<0.001; season 2016–2018: OR 8.9 (95%CI: 5.5–14.5), p<0.001]. However, only eighty-one patients (16.7%) reported vaccination / willingness for

**Table 1. Demographic parameters and influenza vaccination rates in all patient groups.**

|  | All | RTX | CKD | HD | LTX | ACLD |
|---|---|---|---|---|---|---|
| Number of included patients, N (%)/ | 516 (100) | 293 (56.8) | 73 (14.1) | 95 (18.4) | 10 (1.9) | 45 (8.7) |
| Age, mean±SD | 56.4±14.9 | 56.0±13.8 | 51.6±18.0 | 60.4±15.6 | 53.4±11.2 | 58.3±13.5 |
| Female gender, N (%) | 196 (38.0) | 111 (37.9) | 35 (47.9) | 36 (37.9) | 4 (40) | 10 (22.2) |
| Marital status, N (%) |  |  |  |  |  |  |
| Single | 130/514 (25.3) | 69/292 (23.6) | 24/73 (32.9) | 29/94 (30.9) | 3/10 (30) | 5/45 (11.1) |
| Married | 285/514 (55.4) | 175/292 (59.9) | 35/73 (47.9) | 42/94 (44.7) | 6/10 (60) | 27/45 (60.0) |
| Divorced/Widowed | 99/514 (19.3) | 48/292 (16.4) | 14/73 (19.2) | 23/94 (24.5) | 1/10 (10) | 13/45 (28.9) |
| Highest school degree, N (%) |  |  |  |  |  |  |
| No school degree | 10/507 (2.0) | 6/287 (2.1) | 0 | 3/95 (3.2) | 0 | 1/45 (2.2) |
| Mandatory school | 79/507 (15.6) | 47/287 (16.4) | 7/70 (10.0) | 18/95 (18.9) | 0 | 7/45 (15.6) |
| Vocational training/apprenticeship | 224/507 (44.2) | 142/287 (49.5) | 30/70 (42.9) | 33/95 (34.7) | 7/10 (70.0) | 12/45 (26.7) |
| High school degree | 98/507 (19.3) | 41/287 (13.6) | 19/70 (27.1) | 24/95 (25.3) | 1/10 (10.0) | 13/45 (28.9) |
| University or college degree | 79/507 (15.6) | 39/287 (13.3) | 17/70 (18.6) | 17/95 (17.9) | 2/10 (20.0) | 8/45 (17.8) |
| Self-assessed risk*, N (%) |  |  |  |  |  |  |
| Higher risk of influenza infection | 343/470 (73.0) | 225/270 (83.3) | 38/66 (57.6) | 55/88 (62.5) | 4/8 (50.0) | 21/38 (55.3) |
| Higher risk of severe influenza course | 289/398 (72.6) | 193/235 (82.1) | 28/50 (56.0) | 48/75 (64.0) | 5/8 (62.5) | 48/30 (64.0) |
| Influenza vaccination, N (%) |  |  |  |  |  |  |
| Vaccination planned this season | 213/485 (43.9) | 129/272 (47.4) | 30/70 (42.9) | 40/92 (43.5) | 2/10 (20.0) | 12/41 (29.3) |
| Vaccinated last season | 127/500 (25.4) | 83/284 (29.2) | 18/73 (24.7) | 22/91 (24.2) | 1/10 (10.0) | 3/42 (7.1) |
| Vaccinated 2016–2018 | 135/495 (27.3) | 91/279 (32.6) | 18/72 (25.0) | 22/92 (23.9) | 1/10 (10.0) | 3/42 (7.1) |

ACLD: advanced chronic liver disease, CKD: chronic kidney disease, HD: hemodialysis, IQR: interquartile range, LTX: liver transplantation, N: number, RTX: renal transplantation, *self-estimated risk compared to healthy individuals.

vaccination at all three time points. Three (0.6%) patients indicated that they did not yet decide regarding influenza vaccination. Willingness for vaccination was highest in kidney transplant- (47.4%) and lowest in liver transplant recipients (20%). The vaccination willingness did not differ significantly between all five groups (p = 0.25). Patients planning influenza vaccination were older [59.5/50.0–68.0 years vs. 56.0/44.0–66.0 years (median/IQR), p = 0.024] and more often born in Austria (47.8% vs. 30.6%, p = 0.009). Other sociodemographic- [female gender: OR: 1.2 (95%CI: 0.8–1.7), p = 0.442, marital status: married compared to single: OR: 1.1 (95% CI: 0.7–1.8), p = 0.554; divorced/widowed compared to single: OR: 0.9 (95%CI: 0.5–1.5), p = 0.660), level of education (as displayed in Table 2)] and transplant-associated parameters [deceased donor: OR: 0.8 (95%CI: 0.5–1.5), p = 0.482; number of transplantations: OR: 0.8 (95%CI: 0.5–1.4), p = 0.413] did not change the likelihood of vaccination willingness. Ninety-one (18.1%) and 29 (5.9%) patients had received vaccination against streptococcus pneumoniae and haemophilus influenzae, respectively. Older age (per 10 years: OR: 0.9 (95%CI: 0.8–1.2), p = 0.856) was not associated with pneumococcal vaccination status.

## Perceptions of patients on the potential risk of influenza infection and a severe course of disease

The majority of participants believed that their risk of being infected with (n = 343, 73%) and of having influenza-associated complications (n = 289, 72.6%) was higher than the general population. The proportion of patients reporting these concerns was highest in kidney transplant recipients (83.3% and 82.1%) and differed across the groups (patients with CKD not receiving kidney replacement therapy 57.6% and 56.0%, patients receiving hemodialysis 62.5%

**Table 2. Uni- and multivariable regression analysis evaluating factors associated with vaccination willingness.**

| Parameters | Univariable | | Multivariable | |
|---|---|---|---|---|
| | HR (95% CI) | P-value | HR (95% CI) | P-value |
| Age, per 10 years | 1.2 (1.0–1.3) | 0.008 | 0.8 (0.7–1.0) | 0.064 |
| Female | 1.2 (0.8–1.7) | 0.442 | - | |
| Marital status | | | - | |
| Single | 1 | | | |
| Married | 1.1 (0.7–1.8) | 0.554 | | |
| Divorced/Widowed | 0.9 (0.5–1.5) | 0.660 | | |
| Highest school degree | | | - | |
| No school degree | 1 | | | |
| Mandatory school | 2.6 (0.5–13.3) | 0.256 | | |
| Vocational training/apprenticeship | 2.5 (0.5–12.3) | 0.259 | | |
| High school degree | 3.0 (0.6–15.3) | 0.183 | | |
| University/college degree | 3.6 (0.7–18.5) | 0.125 | | |
| Other | 3.5 (0.5–22.3) | 0.185 | | |
| Previous influenza vaccinations | | | | |
| Vaccination 2019/2020 | 17.1 (9.5–30.7) | <0.001 | 20.3 (7.9–52.0) | <0.001 |
| Vaccination 2016–2018 | 8.9 (5.5–14.5) | <0.001 | 3.9 (1.7–9.1) | 0.002 |
| Transplant-associated parameters | | | - | |
| Deceased donor | 0.8 (0.5–1.5) | 0.482 | | |
| Number of transplantations | 0.8 (0.5–1.4) | 0.413 | | |
| Self-assessed risk | | | | |
| Higher risk for influenza infection | 2.8 (1.8–4.5) | <0.001 | 1.5 (0.6–3.6) | 0.364 |
| Higher risk for severe influenza course | 3.8 (2.3–6.3) | <0.001 | 2.4 (1.0–5.8) | 0.046 |
| Change of opinion due to COVID-19 pandemic | | | | |
| Yes | 4.8 (3.0–7.5) | <0.001 | 23.1 (10.9–49.1) | <0.001 |

Uni- and multivariable logistic regression analysis: within the groups marital status and school degree the status "single" and "no school degree" were the reference parameters; CI: confidence interval, HR: hazard ratio, COVID-19: coronavirus disease of 2019.

and 64%, liver transplant recipients 50% and 62.5%, patients with ACLD 55.3% and 50%; p<0.001 for both risks). Declared willingness to receive vaccination was associated with the self-perceived risks of infection (OR: 2.8 95%CI: 1.8–4.5, p<0.001) and associated complications (OR: 3.8 (95%CI: 2.3–6.3), p<0.001).

## Factors supporting influenza vaccination

Of all 127 patients who received influenza vaccination in the last season, 102 (80.3%) patients underwent influenza vaccination because it was recommended by medical personnel (Fig 1). Sixty patients (58.8%) followed the recommendations from their general practitioners, 26 patients (25.5%) from their outpatient clinic specialist, 7 patients (6.9%) from external physicians, 19 patients (18.6%) from their nephrologists in the dialysis unit and 5 patients (4.9%) from hospital nurses (Fig 2). Eleven patients (8.7%) reported that a previous influenza infection was their motivation for influenza vaccination. 45 patients (35.4%) reported that their motivation to be vaccinated was self-acquired knowledge. This specific reason was reported significantly more frequently from patients with high school degree or university degree (15.3% vs. 8.0%, p = 0.012). Patients believing to have an increased risk for influenza infection, were almost three times more likely (OR: 2.8 (95%CI: 1.8–4.5), p<0.001) to be willing to

## Motivation for influenza vaccination

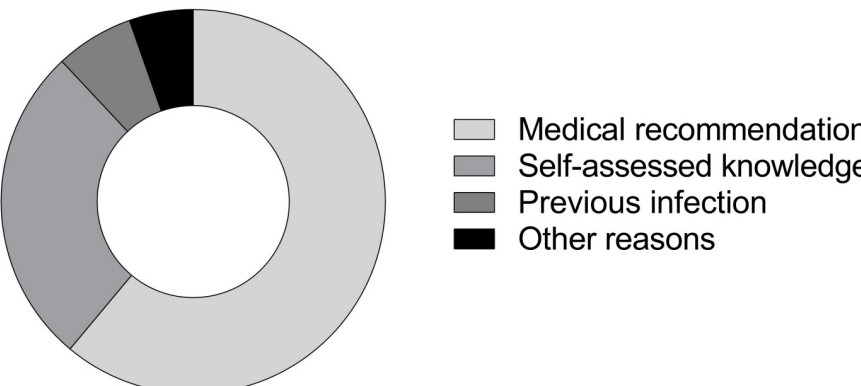

**Fig 1. Motivation for influenza vaccination.** Fig 1 illustrates relative frequencies of reported reasons for influenza vaccination in patients who underwent vaccination in the last season. Relative frequencies: recommendation N = 102/80.2%, self-assessed knowledge N = 45/35.4%, previous infection N = 11/8.7%, other reasons N = 9/7.1%.

receive influenza vaccination. In line, patients stating to have an increased risk for influenza associated complications were four times more likely [OR: 3.8 (95%CI: 2.3–6.3), p<0.001] to accept influenza vaccination.

### Factors negatively associated with influenza vaccination

In total, 373 (74.6%) patients reported that they were not vaccinated in the previous season. The most frequent reason was fear of side effects or complications (N = 121/32.4%, Fig 3). Forty-three (11.5%) patients reported that they experienced adverse events following previous influenza vaccinations. Fifty-six (15%) patients indicated low expectations regarding the success of the vaccination (prevention of disease) and twenty-eight (7.5%) patients reported that the financial costs of influenza vaccination would be a significant obstacle. In the group of

## Source of recommendation

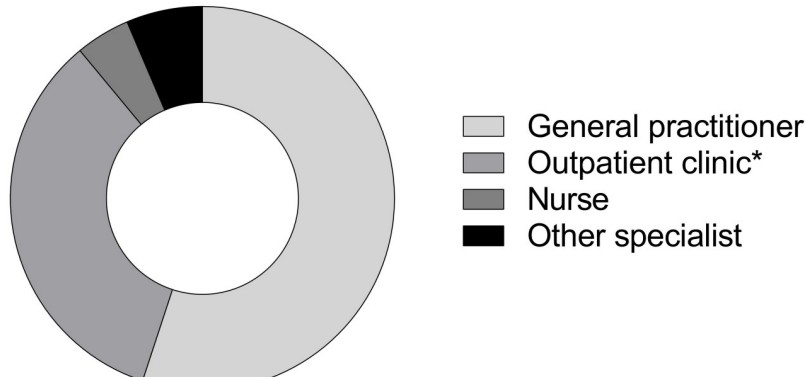

**Fig 2. Source of recommendation in patients undergoing vaccination.** Fig 2 illustrates relative frequencies of reported sources of recommendations for influenza vaccination in patients who underwent vaccination in the last season. Relative frequencies: general practitioner N = 60/58.8%, outpatient clinic (*including dialysis unit) N = 37/36.3%, nurses N = 5/4.9%, external specialist N = 7/6.9%.

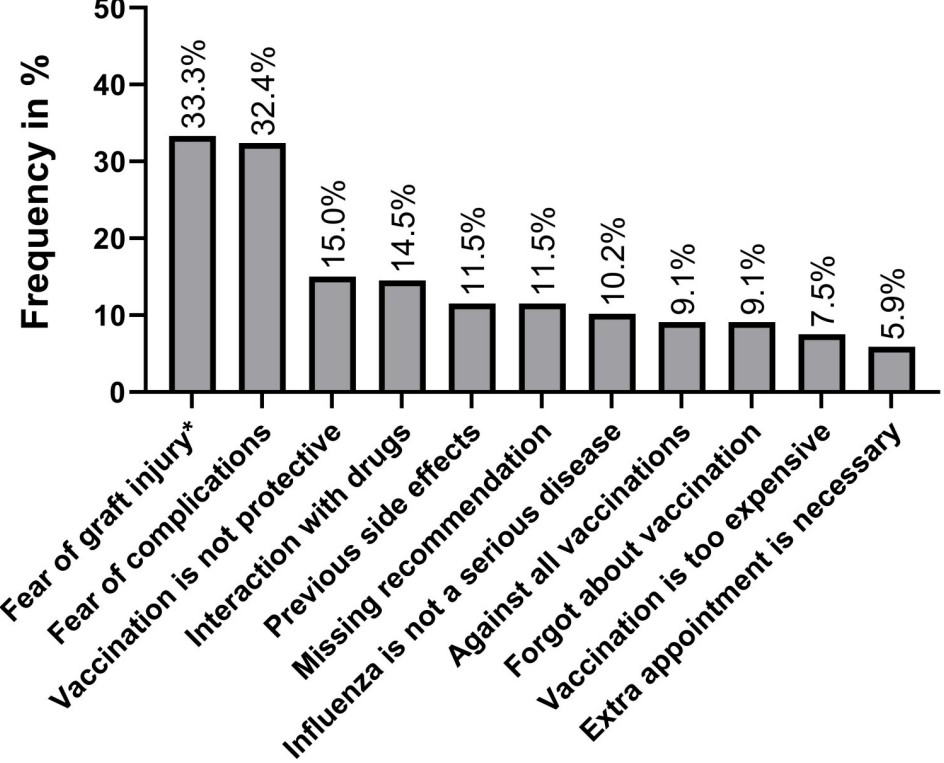

**Fig 3. Most frequent arguments against influenza vaccination.** Fig 3 illustrates most common arguments against influenza vaccination: *only allograft recipients included. Further, not included reasons: family advised against vaccination (N = 17/4.6%), media advised against vaccination (N = 6/1.6%), vaccination is not available (N = 16/4.3%), side effects experienced with other vaccines (N = 11/2.9%).

transplanted patients (both kidney and liver), the most frequent argument against vaccination was the fear of graft injury (33.3%). The frequency of these reasons was associated with the country of origin of the patients. While patients not born in Austria reported significantly more often about financial obstacles (16.3% vs. 4.8%; p<0.001), profound skepticisms regarding the success of the vaccination was significantly more frequent in patients born in Austria (4.3% vs. 18.8%, p = 0.003). Patients who reported financial expenses as reasons against vaccination were significantly younger when compared to those who did not mention the financial burden of vaccination [48/36-60 years vs. 58/48-67 years (median/IQR); p = 0.005].

### Influence of COVID-19 pandemic on the influenza vaccination awareness

135 patients (29.5%) reported that the COVID-19 pandemic changed their subjective opinion towards influenza vaccination: the vast majority (83%) of these patients reported that their personal opinion on influenza vaccination improved throughout COVID-19 pandemic.

### Multivariable analysis of factors associated with the willingness to receive influenza vaccination

Next, we performed a multivariable analysis by including variables with statistically significant results in univariable regression analyses into a multivariable model (Table 2). While age [per

10 years: OR: 0.8 (95%CI: 0.7–1.0), p = 0.064] and a higher self-perceived risk for infection [OR: 1.5 (95%CI: 0.6–3.6), p = 0.364] did not attain statistical significance, previous vaccinations [2019/2020: OR: 20.3 (95%CI: 7.9–52.0), p<0.001 and 2016–2018: OR: 3.9 (95%CI: 1.7–9.1), p = 0.002] as well as a higher risk for a severe course of disease [OR: 2.4 (95%CI: 1.0–5.8), p = 0.046] were independently associated with vaccination willingness. Importantly, a change of opinion due to COVID-19 pandemic resulted in the highest odds ratio for vaccination willingness [OR: 23.1 (95%CI: 10.9–49.1), p<0.001].

## Discussion

We found increasing influenza vaccination willingness for the upcoming 2020/2021 season in all included patient groups. We hypothesize that the COVID-19 pandemic might be the primary reason for the increasing willingness to receive influenza vaccination. Almost 30% of patients believed that influenza vaccination was becoming more important since the outbreak of the pandemic and this factor remained a significant predictor for vaccination willingness in multivariable regression analysis. We also found that patient´s individual risk perception and their previous vaccination behavior were strong indicators for the plan to receive vaccination. Nevertheless, these numbers still indicate an urgent need for improvement. In Germany–representing a country with a comparable health care system to Austria—influenza vaccination rates of 31%-33% in kidney transplant recipients and of 42–44% in patients undergoing haemodialysis have been reported for the time period of 2012 to 2017 [13]. Although vaccination rates in renal transplant recipients were comparable in our study, included hemodialysis patients reported profoundly lower rates in the previous years. This may seem counterintuitive, as hemodialysis patients receive closer medical surveillance due to weekly visits by nephrologists. However, our study revealed that general practitioners and specialists at outpatient clinics were the most important sources of vaccine recommendations. Additionally, the costs for pre-emptive treatments such as vaccinations are not reimbursed by Austrian social insurances in the hospital setting explaining the fact that influenza vaccination in Austria is mostly performed by general practitioners. As hemodialysis patients are mostly managed by medical specialists, primary health care utilization might be underrepresented. In line with this assumption, a recent study confirmed that dialysis patients that are also treated by a primary care physician, were significantly more likely to receive influenza vaccination [24].

Most frequent arguments against influenza vaccination included the fear of side effects and allograft injury. The fear of side effects was also a prominent argument in previous questionnaire studies, even in medical staff [8, 25–27]. The safety of seasonal trivalent inactivated influenza vaccines as well as the high dose and booster dose has been confirmed in several studies including transplant recipients [28, 29]. A large meta-analysis from 2018 also showed no increased risk for the development of donor-specific-antibodies or rejection in influenza vaccinated patients [30]. In contrast, these studies even suggested that the risk of acute cellular rejection and chronic allograft dysfunction is significantly higher for both renal- and liver transplant recipients following influenza infection [31]. Additionally, influenza infection was recently identified as an important trigger for the development of acute-on-chronic liver failure (ACLF), which occurred in almost every fifth patient with liver cirrhosis who had to be hospitalized due to influenza infection. In this study, influenza infection led to the development of organ failures, secondary infections and death underlining the importance of influenza vaccination in this population [14]. In line, a recent meta-analysis including 12 studies evaluating the effectiveness of influenza vaccination in patients with chronic liver disease also concluded that potential benefits outweigh the costs and risks associated with vaccination [32]. Therefore, providing proper patient information on potential negative consequences of

influenza infection may help to increase compliance to vaccination programs in these special populations.

Influenza vaccination rates in the last years were higher compared to those previously reported from the general Austrian population [6] and comparable to those reported in patients with rheumatic diseases in Austria [8]. Influenza vaccination proponents were significantly older as compared to those unwilling to receive vaccination, which is in line with findings reported by Harrison et al. [8]. Vaccination recommendation awareness in younger patients may be lower in primary health facilities as national influenza vaccination recommendations primarily target elderly individuals. Interestingly, patients with migratory background and patients of younger age reported that vaccine-associated costs were a significant reason for refusing influenza vaccination. Before Covid-19, influenza vaccination was not routinely covered by public health insurance in Austria, but currently free of charge vaccination programs were initiated in order to support the public health system. A previous study showed that reimbursement of vaccination expenses can improve vaccination acceptance significantly [33] and further analysis of vaccination rates after this season will be needed to evaluate, if this will also be true during the ongoing COVID-19 pandemic.

Our study has some potential limitations. Patients were asked to complete questionnaires without assistance and due to limited personnel capacity, we were unable to evaluate the return rate of distributed questionnaires. The survey was conducted in German, thus patients may have been precluded due to language barriers. Nevertheless, patients with a significant language barrier, are generally asked to bring a family member or a friend capable of understanding and speaking German or English which might have compensated this possible bias to a certain degree. Furthermore, the main parameter of interest was the influenza vaccinate rate in the upcoming winter season and false statements due to socially expected behavior cannot be excluded. Nevertheless, we tried to minimize this potential effect by completely anonymizing the survey. Additionally, self-assessed vaccination data might not always be the most appropriate vaccination uptake measure, however, due to the anonymous study design, we could not link data derived from the questionnaires to electronic patient records.

In conclusion, our data show increasing influenza vaccination willingness in patients with chronic kidney or liver disease. The fear of complications or graft injury, inefficient vaccinations and missing recommendations were identified as most frequent arguments against vaccination. On the other hand, most patients considered themself as being at greater risk of influenza infection—indicating that vaccination readiness can be significantly improved if disruptive factors are appropriately addressed. Improving co-operations with primary health care providers, active vaccination surveillance programs and specialist consultations including more comprehensive information on potential adverse events or complications and political measures such as financial reimbursement might help to further promote vaccination rates in this endangered patient population.

## Supporting information

**S1 Table. Demographic- and transplant associated parameters and influenza vaccination willingness.**
(DOCX)

## Author Contributions

**Conceptualization:** Michael Eder, Zeljko Kikic, Marcus D. Saemann, Allison Tong, Thomas Reiberger, Heimo Lagler, Bernhard Scheiner.

**Data curation:** Haris Omic, Jana Gorges, Florian Badt, Marcus D. Saemann, David Bauer, Georg Semmler, Bernhard Scheiner.

**Investigation:** Zeljko Kikic, Allison Tong, Heimo Lagler.

**Methodology:** Michael Eder, Allison Tong.

**Project administration:** Michael Eder, Thomas Reiberger, Heimo Lagler.

**Supervision:** Allison Tong.

**Writing – original draft:** Michael Eder, Haris Omic.

**Writing – review & editing:** Michael Eder, Marcus D. Saemann, Allison Tong, Georg Semmler, Thomas Reiberger, Heimo Lagler, Bernhard Scheiner.

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
