## [Decision Letter · Decision Letter 0]

5 Feb 2021

PONE-D-21-00261

Influenza vaccination uptake and factors influencing vaccination decision among patients with chronic kidney or liver disease

PLOS ONE

Dear Dr. Scheiner,

Thank you for submitting your manuscript to PLOS ONE. After careful consideration, we feel that it has merit but does not fully meet PLOS ONE’s publication criteria as it currently stands. Therefore, we invite you to submit a revised version of the manuscript that addresses the points raised during the review process.

**The manuscript focuses on a potential relevant topic. The study, however, presents several shortcomings that should be addressed before reaching sound conclusions. To mention some of them, i) need to elaborate in the Introduction on  the recommendations issued by the Austrian Federal Health Ministry; ii) concern about the fact that the study is mentioned as prospective, but it seems a cross-sectional study; iii) need to clarify how the questionnaires were distributed, anonymized , and returned to the researchers; iv) major concern about the applied statistical tests through the manuscript; v) unclear on the basis of which question the authors have considered someone a vaccination proponent; vi) it would be useful to do a logistic regression for all factors and a multiple logistic regression could as well be considered.**

We look forward to receiving your revised manuscript.

Kind regards,

Giuseppe Remuzzi

Academic Editor

PLOS ONE

Journal Requirements:

2. Our staff editors have determined that your manuscript is likely within the scope of our Call for Papers on Influenza. This editorial initiative is headed by PLOS ONE Guest Editors Dr. Meagan Deming and Dr. Deshayne Fell. The Collection encompasses research on influenza prevention on every level, including in vitro, translational, behavioral, and clinical studies; disease and immunity modelling; as well as new approaches to influenza prevention.

Additional information can be found on our announcement page: https://collections.plos.org/call-for-papers/influenza/

Currently, your manuscript is included in the group of papers being considered for this call. Please note that being considered for the Collection does not require additional peer review beyond the journal’s standard process and will not delay the publication of your manuscript if it is accepted by PLOS ONE.

We would greatly appreciate your confirmation that you would like your manuscript to be considered for this Collection by indicating this in your next cover letter. If you would prefer to remove your manuscript from collection consideration, please specify this in your cover letter.

3. Please include additional information regarding the survey or questionnaire used in the study and ensure that you have provided sufficient details that others could replicate the analyses.

For instance, if you developed the survey or questionnaire as part of this study and it is not under a copyright more restrictive than CC-BY, please include a copy, in both the original language and English, as Supporting Information.

If the questionnaire is published, please provide a citation to the (i) questionnaire and/or (ii) original publication associated with the questionnaire.

5. Please ensure that you refer to Figures 1, 2 and 3 in your text as, if accepted, production will need this reference to link the reader to each figure.

Reviewers' comments:

Reviewer's Responses to Questions

**Comments to the Author**

1. Is the manuscript technically sound, and do the data support the conclusions?

Reviewer #1: Partly

2. Has the statistical analysis been performed appropriately and rigorously? 

Reviewer #1: No

3. Have the authors made all data underlying the findings in their manuscript fully available?

Reviewer #1: Yes

4. Is the manuscript presented in an intelligible fashion and written in standard English?

Reviewer #1: Yes

5. Review Comments to the Author

Reviewer #1: This is a study on a very relevant topic. Statistical methods should, however, be revised.

Minor comments:

Introduction:

1) P5, line 9: is “siloed” the correct word?

2) Could you elaborate on the recommendations issued by Austrian Federal Health Ministry? (e.g. which groups, ages)

Materials and methods:

1) It is described that the study is prospective. However, to me it seems more as a cross-sectional study. Please check.

2) Could you clarify were the questionnaires distributed, how they were anonymized and how were they returned to the researchers?

3) Peritoneal dialysis patients were not included. Could you add the reason for this?

4) If questions were asked about h. influenzae and s. pneumoniae, it would be good to add the Austrian recommendations concerning these pathogens in the introduction.

Results:

1) ACLD is not described equally in the text (Advanced chronic liver disease) and in footnotes of table 1 (acute on chronic liver disease)

2) Could you refer in the text to the figures?

3) Could you add a translation of the questionnaire as supplementary material? This will clarify the methodology for the reader.

Discussion:

1) It is described that there is an increased willingness to vaccinate. However, willingness was only assessed in the current influenza season. Actual vaccination uptake is not the same as willingness to be vaccinated. It would be good to clarify this.

2) It would be good to add in the limitation section that vaccination recall-bias might have played a role and that self-assessed vaccination data might not always be the most appropriate vaccination uptake measure.

Major Comments:

I have concerns about the applied statistical tests, this should be checked and clarified throughout the manuscript

1) It is stated in method section that for comparison of continuous data unpaired Student´s t- or Mann-Whitney-U-test, and for correlation analysis Spearman’s Correlation Coefficient were used.

- Could it be clarified which test was used for what?

2) In results: “Influenza vaccination in the previous season (2018/2019) as well as in the years before (2016-2018) was significantly associated with the intention to receive influenza vaccination this season (R=0.511, p<0.001 and R=0.440, p<0.001).”

Is this a Spearman’s correlation as described in the methods section? Assuming that both the dependent and independent variable are dichotomous, spearman’s correlation might not be an appropriate measure. Spearman’s correlation is a measure for the association between two continuous or ordinal variables and thus not dichotomous variables. Other tests might be more appropriate (e.g. reporting odd ratio via binomial logistic regression). This also counts for assessment of other associations later mentioned in result section. Could this be checked?

3) In results: “When comparing between different cohorts, kidney transplant recipients had the highest rate of influenza vaccination proponents (47.4%) compared to patients with CKD not requiring kidney replacement therapy (42.9%), patients receiving HD (43.5%) and patients with ACLD (29.3%).”

- Did you assess the differences between the cohorts statistically?

- Could you add in the methods on the basis of which question you consider someone a vaccination proponent?

4) In results: “Patients planning influenza vaccination were older [59.5/50.0-68.0 years vs. 56.0/44.0-66.0 years (median/IQR), p=0.024] and more often born in Austria (47.8% vs. 30.6%, p=0.009).”

- Which test did you use for assessment of ‘born in Austria’. This is not a continuous variable and tests for non-continuous variables are not described in method section.

- Maybe it could be checked if it would not be better to do a logistic regression for all factors and a multiple logistic regression could as well be considered.

5) In results “The proportion of patients reporting these concerns was highest in kidney transplant recipients (83.3% and 82.1%) and differed across the groups (patients with CKD not receiving kidney replacement therapy 57.6% and 56.0%, patients receiving hemodialysis 62.5% and 64%, liver transplant recipients 50% and 62.5%, patients with ACLD 55.3% and 50%; p<0.001 for both risks).”

- It is not clear which test was used here and if it is determined where the difference lies.

6) In results: “This specific reason was reported significantly more frequently from patients with high school degree or university degree (15.3% vs. 8.0%, p=0.012).”

- which statistical test is used here?

7) In results: “While patients not born in Austria reported significantly more often about financial obstacles (16.3% vs. 4.8%; p<0.001), profound skepticisms regarding the success of the vaccination was significantly more frequent in patients born in Austria (4.3% vs. 18.8%, p=0.003). Patients who reported financial expenses as reasons against vaccination were significantly younger when compared to those who did not mention the financial burden of vaccination [48/36-60 years vs. 58/48-67 years (median/IQR); p=0.005].”

- which statistical test is used here?

6. PLOS authors have the option to publish the peer review history of their article (what does this mean?). If published, this will include your full peer review and any attached files.

Reviewer #1: No

---

## [Author Response · Author response to Decision Letter 0]

5 Mar 2021

Please find our responses to the reviewer and editor comments in the uploaded Response Letter.

---

## [Decision Letter · Decision Letter 1]

25 Mar 2021

Influenza vaccination uptake and factors influencing vaccination decision among patients with chronic kidney or liver disease

PONE-D-21-00261R1

Dear Dr. Scheiner,

We’re pleased to inform you that your manuscript has been judged scientifically suitable for publication and will be formally accepted for publication once it meets all outstanding technical requirements.

Kind regards,

Giuseppe Remuzzi

Academic Editor

PLOS ONE

Additional Editor Comments (optional):

Reviewers' comments:

Reviewer's Responses to Questions

**Comments to the Author**

1. If the authors have adequately addressed your comments raised in a previous round of review and you feel that this manuscript is now acceptable for publication, you may indicate that here to bypass the “Comments to the Author” section, enter your conflict of interest statement in the “Confidential to Editor” section, and submit your "Accept" recommendation.

Reviewer #1: All comments have been addressed

2. Is the manuscript technically sound, and do the data support the conclusions?

Reviewer #1: Yes

3. Has the statistical analysis been performed appropriately and rigorously? 

Reviewer #1: Yes

4. Have the authors made all data underlying the findings in their manuscript fully available?

Reviewer #1: Yes

5. Is the manuscript presented in an intelligible fashion and written in standard English?

Reviewer #1: Yes

6. Review Comments to the Author

Reviewer #1: The authors have adapted the manuscript well according to the comments. I believe the manuscript has improved and is now suitable for publication.

7. PLOS authors have the option to publish the peer review history of their article (what does this mean?). If published, this will include your full peer review and any attached files.

Reviewer #1: **Yes: **Lise Boey

---

## [Editor Report · Acceptance letter]

29 Mar 2021

PONE-D-21-00261R1 

Influenza vaccination uptake and factors influencing vaccination decision among patients with chronic kidney or liver disease 

Dear Dr. Scheiner:

I'm pleased to inform you that your manuscript has been deemed suitable for publication in PLOS ONE. Congratulations! Your manuscript is now with our production department. 

Kind regards, 

on behalf of

Prof. Giuseppe Remuzzi 

Academic Editor

PLOS ONE